# Pharmacovigilance Strategies to Address Resistance to Antibiotics and Inappropriate Use—A Narrative Review

**DOI:** 10.3390/antibiotics13050457

**Published:** 2024-05-16

**Authors:** Valcieny Sandes, Albert Figueras, Elisangela Costa Lima

**Affiliations:** 1Postgraduate Program in Pharmaceutical Sciences, School of Pharmacy, Federal University of Rio de Janeiro, Av. Carlos Chagas Filho-373, Rio de Janeiro 21941-170, RJ, Brazil; eclima.ufrj@gmail.com; 2National Cancer Institute, Pr. da Cruz Vermelha-23, Rio de Janeiro 20230-130, RJ, Brazil; 3Independent Researcher, Santos 11045-101, SP, Brazil; albert.figueras@gmail.com

**Keywords:** pharmacovigilance, antibiotic use, antimicrobial resistance, antimicrobial stewardship, inappropriate use

## Abstract

The spread of antimicrobial resistance (AMR) is a global challenge. Close and continuous surveillance for quick detection of AMR can be difficult, especially in remote places. This narrative review focuses on the contributions of pharmacovigilance (PV) as an auxiliary tool for identifying and monitoring the ineffectiveness, resistance, and inappropriate use of antibiotics (ABs). The terms “drug ineffective”, “therapeutic failure”, “drug resistance”, “pathogen resistance”, and “multidrug resistance” were found in PV databases and dictionaries, denoting ineffectiveness. These terms cover a range of problems that should be better investigated because they are useful in warning about possible causes of AMR. “Medication errors”, especially those related to dose and indication, and “Off-label use” are highlighted in the literature, suggesting inappropriate use of ABs. Hence, the included studies show that the terms of interest related to AMR and use are not only present but frequent in PV surveillance programs. This review illustrates the feasibility of using PV as a complementary tool for antimicrobial stewardship activities, especially in scenarios where other resources are scarce.

## 1. Introduction

Despite the many lives saved after the discovery of antibiotics (ABs), exposure to these drugs also favours the advancement of bacterial resistance. The excessive use of ABs in veterinary and agricultural settings, as well as their environmental presence, leads to the development and spread of resistance genes. In addition to this panorama, another well-known and highly relevant driver of antimicrobial resistance (AMR) is AB misuse in clinical settings (inappropriate prescribing, self-medication) [1,2,3,4,5,6,7,8].

The challenges in overcoming the global progression of AMR through the promotion of an accurate selection and adherence to the guidelines include the following: (i) problems with the standardisation of data and monitoring tools; (ii) difficulties accessing adequate care for diagnosis and treatment; and (iii) an increase in infections caused by resistant pathogens and the need for more effective treatments [9]. Additionally, dispensing without a prescription is a widespread practice that is difficult to control [10]. Low- and middle-income countries (LMICs) are most affected, especially with regard to poor investments in structure, specialized laboratories, and trained personnel for the diagnosis, monitoring, and evaluation (M&E) of AMR [11].

The World Health Organization Global Antimicrobial Resistance and Use Surveillance System was released in 2015 to try to fill gaps in coping with AMR [12]. In 2021, of the 216 countries, territories, and areas in the world, only 55 (25%) provided data on antimicrobial consumption, and 87 (40%) provided bacterial identification results [13]. Since the main cause of AMR is the inappropriate use of ABs, it is equally important to know (i) the prescription profile of these drugs; (ii) the degree of adherence to the treatment guidelines; and (iii) the pattern of nonprescription dispensing, as well as other situations that contribute to both excessive use and irrational consumption. To achieve this goal, monitoring and evaluation (M&E) of antibiotic use at the national level, along with studies on antibiotic utilization across various levels of the supply chain, are recommended. These studies should provide local data to inform policy-making and promote changes in prescribing habits [12,14].

The World Health Organization (WHO) developed the Access, Watch, and Reserve (AWaRE) classification to facilitate the collection and comparison of data for the surveillance of AB consumption and support antibiotic stewardship (AMS) efforts. This classification is based on the impact of various antibiotics and antibiotic classes on antimicrobial resistance. The goal established by the WHO is for 60% of ABs consumed in a country to be by those considered to have a lower risk of AMR (the Access category) [15].

The consumption of ABs varies—between and within countries and regions—and is influenced by cultural, economic, and political factors [16,17]. A recent review indicated that the general increase in the consumption of ABs in LMICs, especially those in the Watch category, worsened during the COVID-19 pandemic [18]. Consistently and uniformly evaluating antimicrobial consumption is a challenge in many countries, such as those in Latin America [19] and Africa [20]. Combined efforts in the most diverse areas of science are necessary, as resistance mechanisms are known for all marketed ABs, and the discovery of promising new classes to treat bacterial infections is increasingly rare [21].

The guide for local policies of integrated AMS, put together by the WHO, is based on five pillars: developing national coordination; ensuring access to and regulation of antimicrobials; improving awareness, education, and training; strengthening infection control through efforts focussed on water supply, sanitation, and hygiene; and surveillance, monitoring, and evaluation [22]. This final pillar includes monitoring how antibiotics are utilised. However, close and continuous surveillance for quick detection of inappropriate use and the appearance of AMR can be difficult, especially because well-equipped laboratories with trained personnel can be scarce in certain settings [9,23]. Hence, another helpful and inexpensive strategy has been suggested for challenging scenarios: the use of pharmacovigilance (PV) data to report suspected cases of AMR, therapeutic failure, and inappropriate or off-label use of ABs [24,25,26,27,28,29].

The WHO definition of PV is “the science and the activities related to the detection, assessment, understanding, and prevention of adverse effects or any other medicine or vaccine-related problem” [30,31]. Thus, by definition, PV is intricately aligned with the pillars of AMS concerning the monitoring, identification, and quantification of risks associated with using ABs. PV features, such as detecting adverse drug reactions (ADRs) and inappropriate use of medicines, have been well-documented in the literature [32]. Signal detection is the backbone of PV, leading to changes in the authorisation status and even withdrawal of medicines due to ADRs—for example, labelling modifications to limit the use of fluoroquinolones [33]. More than 40 percent of the antibiotics released between 1980 and 2009 were withdrawn from the market for safety concerns or lack of effectiveness compared to existing medicines [34], and PV has contributed to this. Thus, this narrative review aims to present an overview of the contributions of PV as a tool for identifying and monitoring suspected resistance and inappropriate use of antibiotics. We synthesized studies that illustrated how PV strategies can be useful in (i) the timely identification of possible AMR clusters that guide authorities for specific testing and (ii) contributing to monitor AB use.

## 2. The Role of PV

Beginning in the 1960s, after the thalidomide tragedy, the concepts of ADRs and PV were established and refined [35]. For more than 50 years, the WHO has contributed to the creation of PV systems based on the reporting of suspected ADRs, with the objective of quickly identifying signals of unknown reactions and preventing new disasters such as those caused by thalidomide. Today, the WHO Program for International Drug Monitoring (WHO-PIDM) has been established in more than 170 countries [36]. Due to this network structure of countries covering 99% of the world population, the WHO-PIDM is the most extensive global network linking healthcare professionals and patients, who can report any suspicion related to medicine use and safety through a simple, quick system that reaches the WHO-PIDM database [37].

PV activities consist of monitoring “events”, their collection, and research to detect any causal relationship between the medicine and the observed harm, i.e., an adverse “reaction”, defined as “a harmful effect suspected to be caused by a medicine” [38]. PV is based on the monitoring of large populations through the reports of multiple observers (health professionals, patients, and manufacturers). This same information allows the evaluation of the use of the involved medicines in different situations, an insight into the behavioural/epidemiological changes that populations are experiencing, and the indirect evaluation of the observed risks and benefits [26,39]. Data from spontaneous reports are key sources for the identification of safety signals related to the use of medicines after they have been approved and marketed, i.e., their use in real life [40,41,42].

One feature of spontaneous reporting is the use of internationally standardised terminologies, which allows for better evaluation and comparison between different countries/regions [40,43]. In addition to the adverse effect itself, the information collected in the standard and simple report form lets us recognise the reason for using the medicine, medicines taken concomitantly, and the underlying diseases of patients affected by an ADR.

Like any observational study, research based on PV reports has limitations, namely, potential biases due to incomplete data, errors in the classification/coding of adverse events (AEs), and especially underreporting [44,45,46]. Still, PV is a useful method for quickly collecting reports of suspected events that, when they reach a critical number, become a signal that can be further investigated in detail with additional methodologically sound studies when needed.

As the field of PV has evolved, large databases have been built at the national (for example, the Food and Drug Administration Adverse Event Reporting System (FAERS) in the United States), regional (for example, EudraVigilance), and global levels (VigiBase, in WHO-PIDM).

In the specific field of ABs, the literature is rich in PV studies that evaluated the profile of AEs of these medicines. These studies aimed to identify the main substances related to a given event (or the reverse, the main events related to an AB), the frequency of observation of an AE/AB pair, and the evaluation of rare events or drug–drug interactions involving an AB, in addition to the identification of populations at risk. Appendix A lists recent studies that exemplify the recognized goals of PV [40,41,42,47,48,49,50,51,52,53,54,55,56,57,58,59,60,61,62,63,64,65,66,67,68,69,70,71,72,73,74,75,76,77,78,79,80,81,82,83,84,85,86,87,88,89,90,91,92].

## 3. PV Strategies Useful for Monitoring the Use of ABs and Combating AMR

ABs are one of the drug classes most frequently associated with AEs [93,94,95,96,97], especially those in the Watch category [18,29,93]. Interestingly, in addition to the adverse reactions themselves (for example, nausea, vomiting, diarrhoea, hypersensitivity, headache, etc.), the dictionaries that describe AEs include terms useful for identifying aspects related to their use. Thus, events such as “lack of the expected therapeutic effect”, “lack of efficacy”, “inappropriate use”, “use out of indication”, and even “suspicion of resistance to ABs” are already coded as Medical Dictionary for Regulatory Activities (MedDRA) terms [29,98]. This terminology is internationally standardised and widely used in health device information systems and PV databases worldwide [99]. Some researchers have noted that the identification of reports involving these terms in PV databases could be used as a proxy for inappropriate use or suspicion of resistance in the absence of specific methods for the detection of these problems [24,25,27,29,98]. Figure 1 depicts the evolution of publications in the literature regarding this proposal.

A pilot study showed the feasibility of finding treatment failures due to AMR in reports of ADRs in VigiBase by identifying a cluster of terms related to the lack of expected therapeutic effect; they were ranked as the ninth most frequently reported adverse reaction [98]. Another study showed how PV data may be relevant in the monitoring of AMR by relating data from suspected AE reports to ABs used to treat carbapenemase-producing *Klebsiella pneumoniae* (KPC) infections in three different databases in two countries with high (Italy) and low prevalence of KPC (United Kingdom), based on data on culture isolates provided by the European Centre for Disease Prevention and Control. The positive correlations showed an overall increase in AEs, with emphasis on these terms related to AMR: “drug ineffective”, drug resistance”, “pathogen resistance”, “off-label use”, and “product use issue”. They also showed serious AEs related to KPC outbreaks in both countries, showing that it is possible to identify AMR outbreaks earlier by monitoring the evolution of AE notifications related to ABs [26].

It is important to emphasize that the objective of PV is always to generate suspicions of AEs. When analysing a large number of suspicions related to the same medication in a specific location/region, these similar suspicions (using various terms that may lead to the same type of AE) constitute a cluster of reports that generate an alert or signal. In other words, the importance of PV lies precisely in analysing the suspicions and understanding what lies behind them, which could be resistance or inappropriate use.

Figure 2 presents a scheme of the connections between the factors that influence the use of antibiotics in real life and their relationship with AMR that can be identified in PV studies.

## 4. PV as a Tool to Identify the Ineffectiveness of Antibiotics

There are many reasons why a medicine may not achieve its goal. Some of these issues are related to incidents such as the prescription of an inadequate dose, a choice of AB that disregards the clinical guidelines, or nonadherence to treatment. Other causes include drug–drug interactions, comorbidities (e.g., pharmacokinetic changes in critically ill patients), the nutritional status of the user, and the use of counterfeit drugs or drugs with low doses of the active ingredient in addition to biopharmaceutical problems [100,101,102,103,104,105,106,107].

Previous research highlighted specific MedDRA terms in PV databases denoting lack of effect, such as “drug ineffective”, “treatment failure”, “decreased drug efficacy”, and “therapy nonresponder” [29]. Monitoring signs related to these events can be crucial in the investigation of their causes. In addition, AEs related to ineffectiveness may also be a trigger to suspect resistance to the involved AB [28]. The following summarises the findings related to these events and ABs described in PV systems in several countries.

A comprehensive VigiBase analysis (1968–2018) of AB-related AEs (*n* = 1,170,751) found 15,250 (1.3%) reports that reported 17 MedDRA preferred terms (PT) of interest for AMR. The six most reported medicines, which corresponded to 38% of these notifications (*n* = 5806), were amoxicillin (*n* = 873; 5.7%; Access), cephalothin (*n* = 151; 1%; Access), ciprofloxacin (*n* = 1748; 11.5%; Watch), clarithromycin (*n* = 991; 6.5%; Watch), levofloxacin (*n* = 1342; 8.8%; Watch), and daptomycin (*n* = 701; 4.6%; Reserve). The terms most frequently reported in these notifications were “drug ineffective” (*n* = 6959; 45.6%), “off-label use” (*n* = 1455; 9.5%), and “pathogen resistance” (*n* = 1327; 9.0%) [29].

In the Netherlands PV centre, 252 notifications were identified between 1998 and 2019 with MedDRA PTs considered relevant for AMR. The frequencies of “off-label” (*n* = 91; 36.1%), “drug ineffective” (*n* = 71; 28.2%), “product use in inappropriate indications” (*n* = 28; 11, 1%), “pathogen resistance” (*n* = 14; 5.6%), and “drug resistance” (*n* = 13; 5.2%) were detected. That study also described the involved ABs: tobramycin (*n* = 89; 35%; Watch), colistin (*n* = 30; 11.9%; Reserve), ciprofloxacin (*n* = 16; 6.3%; Watch), doxycycline (*n* = 14; 5.5%; Access), and aztreonam (*n* = 12; 4.8%; Reserve) [28].

A study conducted in the European PV database (Eudravigilance—through 2022) described the relationship between MedDRA terms that indicated “drug resistance” and “drug ineffectiveness” and three ABs commonly used in critically ill patients: meropenem (*n* = 8864; Watch), colistin (*n* = 983; Reserve), and linezolid (*n* = 13,381; Reserve). The terms associated with “drug resistance” were more frequent for colistin (*n* = 83; 8.4%) than for meropenem (*n* = 316; 3.6%) or linezolid (*n* = 319; 2.4%). Colistin was also the one most related to “drug ineffectiveness” (*n* = 100; 10.1%), followed by meropenem (*n* = 838; 9.5%) and linezolid (*n* = 556; 4.2%). Suspected bacterial resistance was related to the outcome of death in 20% of the records associated with meropenem, 24% with colistin, and 6% with linezolid. In the “drug ineffective” group, death was reported in 28%, 35%, and 19% of the records, respectively. The authors of that study also suggested that colistin was more likely to be associated with reports of events related to resistance and inefficacy than any ABs other than ceftazidime/avibactam [39].

One study in the Portuguese database described 59,022 reports (2017–2019) related to the most widely used antibiotics in the country for treating upper respiratory tract infections, both in outpatient and hospital settings. The term “drug ineffective” was among the 10 most frequently found MedDRA PTs related to these ABs, with greater emphasis on amoxicillin (*n* = 105; 0.18%; Access), amoxicillin/clavulanic acid (*n* = 295; 0.5%; Access), azithromycin (*n* = 206; 0.35%; Watch), cefazolin (*n* = 30; 0.05%; Access), ciprofloxacin (*n* = 299; 0.51%; Watch), clarithromycin (*n* = 140; 0.24%; Watch), and levofloxacin (*n* = 220; 0.37%; Watch) [108].

In an analysis of 1722 notifications of AEs involving amoxicillin from 1988 to 2014 in the Korea Adverse Event Reporting System database, the third most reported AE was “drug ineffective” (*n* = 174; 10%). The authors attributed the ineffectiveness of amoxicillin to the failure of patients to take the drug for the prescribed time [109].

In Brazil, 12,665 cases of AEs reported between 2018 and 2021 in VigiMed, the Brazilian PV database, were analysed, and the main drugs reported were vancomycin (*n* = 1733; 14%; Watch), ceftriaxone (*n* = 1274; 10%; Watch), piperacillin/tazobactam (*n* = 1024; 8%; Watch), ciprofloxacin (*n* = 936; 7%; Watch), and azithromycin (*n* = 870; 7%; Watch). The event “drug ineffective” was identified as a safety signal associated with azithromycin (*n* = 8; 1%), cefuroxime (*n* = 7; 3.3%), amoxicillin (*n* = 4; 1.3%), and amoxicillin/clavulanic acid (*n* = 8; 1.8%) [110].

Mhaidat et al. evaluated 539 notifications from Jordan (2003–2022). The most frequently reported ABs were tetracyclines (*n* = 101; 19%), fluoroquinolones (*n* = 54; 10%), third-generation cephalosporins (*n* = 48; 9%), and carbapenems (*n* = 42; 8%). In 279 reports, it was possible to characterise the AEs, and the “drug ineffective” MedDRA PT (*n* = 28; 5.19%) was most often related to carbapenems (*n* = 9; 32%), followed by tetracyclines (*n* = 8; 28%). The study also found “off-label” reports (*n* = 13; 2.41%), “incorrect route of administration” (*n* = 10; 1.89%), and “unapproved indication” (*n* = 6; 1.11%) [111].

In India, 1980 reports of suspected AEs were identified and evaluated for causality at a PV monitoring centre (2013–2019). Antimicrobials drew attention as the class most involved in suspicions (29%). Among the relevant AEs found, “multidrug resistance” was related to ceftriaxone/tazobactam (*n* = 15; 0.76%; the Not Recommended category) and amikacin (*n* = 9; 0.45%; Access) [112].

An investigation of tigecycline-related AEs reported to FAERS between 2004 and 2009 found 1182 occurrences. The event “drug ineffective” emerged as one of the most reported (*n* = 63; 5.33%), along with “pathogen resistance” (*n* = 22; 1.86%) [113]. Another study done on the same database (2015–2018) found 5899 notifications related to carbapenems, and again, “drug ineffective” was the most prevalent event (*n* = 620; 10.51%) [66].

These summarised studies demonstrate how spontaneous notifications of suspected ADRs can indicate ineffectiveness and potential antimicrobial resistance. With these data, specific notifications can be investigated to understand the underlying cause of the reported adverse event.

## 5. PV as a Tool to Identify Inappropriate Uses of Antibiotics

In addition to the consumed amount of AB, the quality of consumption in terms of appropriateness of the selection and adherence to clinical recommendations can provide valuable information because, as already mentioned, the inappropriate use of ABs is one of the main causes of AMR. In Africa, for example, a continent highly affected by infectious diseases, high consumption and inappropriate use of ABs were associated with low rates of adherence to prescribing guidelines, high rates of prolonged AB prophylaxis and/or prophylaxis with more than one AB, and high consumption of ABs in the Watch group [114,115,116].

In the National PV Database of Vietnam, 6385 notifications related to ABs were found, totalling 11,652 AE/AB pairs in a 1-year period (2018–2019). Among the ten most reported suspected ABs, eight are classified in the Watch group, namely cefotaxime (*n* = 1142; 17.9%), ceftriaxone (*n* = 395; 9.9%), ceftazidime (*n* = 394; 6.2%), cefoperazone (*n* = 293; 4.6%), cefuroxime (*n* = 242; 3.8%), ciprofloxacin (*n* = 651; 10.2%), levofloxacin (*n* = 287; 4.5%), and vancomycin (*n* = 306; 4.8%), while two belong to the Access group, amoxicillin (*n* = 491; 7.7%) and ampicillin (*n* = 261; 4.1%). Serious events accounted for 49% of the AEs. A total of 889 AEs were considered preventable, 13.4% of which were related to amoxicillin, 10.2% to cefotaxime, and 7.4% to ciprofloxacin. Interestingly, the main causes of these preventable AEs were related to prescribing, which included “inappropriate prescribing” (*n* = 352; 40%), “indicating antibiotics for patients with no sign of infection” (*n* = 272; 31%), “inappropriate indication for antimicrobial prophylaxis” (*n* = 49; 6%), “inappropriate indication” (*n* = 31; 4%), “readministration of antibiotics causing prior allergy/allergies” (*n* = 99; 11%), and “inappropriate dosing” (*n* = 26; 3%). These results reveal evidence of AB overuse that the authors say is related to problems such as difficulty in making an adequate diagnosis, high infection pressure, and problems related to prescribers (personal experience and financial interests) [97].

Ballon et al. conducted a study in the French PV Database (2010–2019) comparing the profile of AEs in pregnant (*n* = 911) and nonpregnant women (*n* = 3358) and reported that anti-infective medicines (*n* = 141; 18.3%) were among the drugs most often associated with these events. The AEs grouped into “lesions, poisoning, and surgical complications”, represented mainly by the MedDRA PTs associated with medication errors, were more frequent among pregnant women (*n* = 71; 8.1%) than among nonpregnant women (*n* = 55; 1.6%) [117].

The Russian PV database (2012–2014) included 3608 reports of beta-lactams: penicillins (*n* = 1123; 31%), cephalosporins (*n* = 2324; 64%), and carbapenems (*n* = 161; 4%). The experts analysed the narrative descriptions of each patient and identified prescription problems and medication errors in 1043 reports (28.9%). Notably, only 29 of these notifications described these problems in the AE field. *n* = Children (*n* = 457; 43.8%) were the age group most involved in this study. The most frequent prescription problems were “administration of an antibiotic in the absence of indications/wrong indication” (*n* = 395; 32.5%) and “contraindicated administration” (*n* = 210; 17.3%); medication errors were “inappropriate dose or dose regimen” (*n* = 360; 29.7%) and “incorrect preparation of an antibiotic solution” (*n* = 27; 2.2%). A relationship with too-low dose frequency was found among the eight (0.2%) patients who experienced treatment failure. Other types of reported prescription problems that deserve attention were “late discontinuation of a drug in a patient developing an ADR” (*n* = 80; 6.6%), “irrational switch of an antibiotic” (*n* = 59 4.9%), “late or irrational change of antibiotic in case of treatment failure” (*n* = 31; 2.6%), and “administration of an inadequate treatment regimen for the disease/treatment strategy” (*n* = 11; 0.9%) [118].

The study conducted in India, described in the previous section, which related “multidrug resistance” to ceftriaxone/tazobactam and amikacin, also called attention to evidence of inappropriate use, since most indications were for empirical prophylaxis, and susceptibility tests for ABs were requested only after treatment failure of the ABs initially prescribed. The observed resistance to ceftriaxone/tazobactam led to substitution with meropenem (Watch) or linezolid (Reserve) [112].

Ren et al. analysed the safety of cephalosporins in two PV databases in China (2009–2010) and reported 1337 AEs. Misuse, especially an “inappropriate dosing regimen”, was observed in 93% of the patients (*n* = 1243), and overuse was observed in 18% (*n* = 249), the latter being more impactful among children [119].

Looking further back in the literature, some studies cited issues related to the misuse of ABs. According to the Iranian PV database (1998–2009), 15.2% (*n* = 183) of the reports for ceftriaxone (*n* = 1205; Watch) were considered preventable [120]. In another study, reports received by the Italian Interregional Group of PV (1988–2004) suggested a higher reporting rate for amoxicillin/clavulanic acid (measured by the number of reports/1,000,000 defined daily doses/year) compared to amoxicillin (2.11 versus 1.52, in the period 200–2004) and an indiscriminate increase in amoxicillin/clavulanic acid, although there are guidelines to make this the second priority [121].

In this section, the included studies demonstrate how certain reported adverse event terms can lead to the identification of antibiotic overuse and misuse.

### Off-Label Use of Antibiotics

Off-label use is not necessarily inappropriate; in certain cases of complicated patients with resistant or uncommon bacteria, empiric use of AB is possible, but always under the close supervision of microbiologists. However, AB treatment without a registered indication and disconnected from evidence of effectiveness against a particular microbial agent, population, and site of action is frequent. Children, pregnant women, and the elderly are populations rarely included in preapproval clinical trials of drugs [122,123,124,125]. Despite this, medicines are often used in these circumstances, just extrapolating the indications and dosage from adult patients without comorbidities, changes in metabolism, or development that would influence pharmacokinetics and pharmacodynamics.

From data in the Swedish database, a study concluded that off-label medicines are more subject to AEs than medicines used as indicated. Furthermore, the paediatric population is more affected than adults [126]. Off-label prescriptions are frequent in children and are more strongly related to severe AEs [127,128].

In Germany, 20,854 reports (2000–2019) were evaluated in the 0–17-year age group, and 3.5% of the respondents had records of off-label use [129]. In Brazil, a study that evaluated 3330 notifications for children up to 12 years of age found that 30% of the reports of fatal cases described an off-label use (*n* = 22). In that study, anti-infective drugs were the most reported class (*n* = 1602; 41%) [130].

An analysis of AEs associated with oxazolidinones reported to FAERS (2018–2020) revealed a more critical safety profile for tedizolid (the newest antibiotic on the market) than expected for serious events such as myelotoxicity, lactic acidosis, and peripheral neuropathy. The authors associated the findings with frequent off-label use, especially regarding the use of this AB for up to 3 weeks, which was longer than what was initially approved (only 6 days) [41].

These studies exemplify some impacts of off-label use of antibiotics that can be identified and monitored through PV activities.

Table 1 summarises the most relevant MedDRA terms and AWaRe categories of associated ABs cited in the studies included in this review.

## 6. Discussion

The revised studies reinforce the potential added utility of the already existing PV systems to collaborate in AMS initiatives by pointing to inappropriate use of ABs and suggesting the existence of possible AMR in remote places with difficult access to laboratory testing and cultures. As already suggested in the literature [24,25,26,29,131] this can be achieved through the promotion of reporting by using the MedDRA PTs of interest for AMR. These studies show how an in-depth analysis of the available data already included in PV databases can provide useful elements for AMS teams at the local or national level and help define antibiotic policies in the country.

In a regional database (Eudravigilance), a higher probability of notification of an AB (in the cited example, colistin) with terms related to resistance and inefficacy was observed than for other antibiotics [39]. Studies such as this one may help doctors choose an AB with a lower probability of resistance/ineffectiveness in a given local reality; these findings can help developing AMS. According to a Korean study [109], the suspicion that the ineffectiveness of amoxicillin was due to inadequate treatment duration shows that encouraging the identification and notification of specific events of interest for in-depth investigations of the causes of ineffective drugs may help us improve the use of ABs. These are some practical examples of the use of the information provided by PV.

The reviewed literature suggests that PV actions could reveal possible AMR and help AMS where lab facilities are scarce. By analysing reports indexed with terms of interest, it becomes possible not only to identify possible resistance but also to gain insight into the utilization patterns of these medications. Report details can provide insights into overuse or misuse that may occur locally or in specific regions/countries. Thus, this information can furnish valuable inputs to antimicrobial stewardship teams and guide strategies for improving antibiotic utilization.

Terms such as “drug ineffective”, “therapeutic failure”, “drug resistance”, “pathogen resistance”, and “multidrug resistance” in PV databases encompass a range of problems that should be better investigated and proven useful in warning about possible causes of AMR before they are used in studies aiming to clarify the contributing factors to be considered in measures to contain AMR. Table 1 reinforces the greater involvement of the Watch category in problems of use and suspicion of AMR and the relevance of the terms listed in identifying these problems.

Although the off-label use of ABs to treat infections caused by resistant pathogens has shown positive results in some situations, the cost of such use must be weighed carefully [132], and it should always be guided by a microbiologist or an infectologist. The studies summarised here emphasise that the term “off-label”, in addition to issues of safety and vulnerability of at-risk populations, frequently appears in PV databases and permeates events of interest to AMR.

Weaknesses in antibiotic therapy and treatment failures related to inadequate dose regimens can also be identified, as exemplified in studies in Russia and China [118,119]. For example, some of the PV studies included in this review suggest the need to review local protocols to ensure that they follow the latest available evidence and the local patterns of AMR, as well as the pharmacokinetics and pharmacodynamics of each AB. Different studies in this revision describe the use of inappropriate doses in treatment failure and bacterial resistance [101,105,133,134]. In other words, the analysis of data included in PV reports may provide clues about inappropriate use in a given place, which allows the design of interventions to improve or reverse this use.

Preventing AEs is an important part of the safe and appropriate use of medicines; furthermore, in the case of ABs, more appropriate use can help contain AMR. On the one hand, there are inappropriate or irrational prescriptions (for example, one of the commonest practices in primary health care: using an AB for mild viral upper respiratory conditions). On the other hand, there are medication errors caused by mistakes or confusion in names, dosages, or duration of treatments. Both cases lead to preventable situations that directly increase the risk of AMR. Concerning medication errors, there is a fear on the part of professionals to record errors in notifications, as demonstrated in the Russian study in which only 2.8% of problems identified were notified [118]. Institutions with higher patient safety culture scores, which report and treat errors, have lower rates of healthcare events [135]. Concerning irrational prescription, it is wider than the prescription of ABs, and it involves many factors beyond inappropriate or outdated knowledge, as psychological, economic, and political aspects also take part in it, together with the characteristics of the healthcare system structure. Still, PV systems have proven useful for indicating points of improvement in the use of health care despite the incompleteness of the records and the underreporting.

Studies to monitor the use of ABs within the scope of PV, prepared according to quality criteria [46,136], can identify a range of associated problems, such as self-medication practices, common practices of inadequate prescriptions and what may influence them, the most prevalent errors, and the ABs most associated with problems of use. The identification of these issues makes it possible to reflect on two features of PV: (i) the regulatory aspect, for which measures to control the use and change the leaflet, policies, and protocols for the use of AB can be established; and (ii) the aspect of the care process, which guides process-improvement actions to minimize medication errors and the development of education programs for health professionals and the population.

Given its structure at the local, national, and global levels, the use of PV systems can be a useful part of strategies to contain AMR, obviously not as a substitute for microbiological laboratory testing but as a system for identifying potential problems of antibiotic efficacy in certain places. Once reported, health authorities and national antimicrobial surveillance teams will have a basis to design the appropriate actions to either obtain microbiological confirmation of the suspicion in the case of AMR or design specific studies to reverse the inappropriate use of antibiotics by using local real-world data. Local/national analyses provide more detailed data reflecting the local epidemiological reality, along with higher-powered application of timely and effective intervention strategies.

Another advantage of PV analysis is that it can be performed over time series derived from local systems and may reflect important changes in use while signalling attention to certain classes of AB. Therapeutic failure analyses may point to the possibility of counterfeiting and substandard products, which are common practices in LMICs and cause great harm to public health [102,137,138]. In addition to the low cost of PV studies, their structure is already widespread and based on international guidelines, and there are PV practices based on active searches, so studies have focussed on specific MedDRA PTs and utilization [139].

It is known that PV programs face challenges, and to fully harness the potential of PV to help AMR issues, it is necessary to overcome some well-known limitations. The main one is underreporting, partly due to a lack of awareness of the importance of reporting or insufficient time and/or human resources for this activity. Additionally, there is significant underreporting because segments of the population and healthcare professionals are unaware of what is reportable and how to report it. Another important issue is the incompleteness of reports or incorrect classifications. It is not a simple task to input AE data into PV systems. Training should be provided to healthcare professionals involved, along with awareness campaigns for the general population. Establishing or encouraging a reporting program for terms of interest related to AMR can set the framework for a win-win relationship. On the one hand, professionals working to combat AMR would benefit from some of these notifications as alerts that something is happening. On the other hand, PV professionals can benefit because more healthcare professionals will be aware of PV’s work and how to report, which would increase notifications overall, strengthening the country’s system.

A limitation of this review is that, due to the method chosen, some studies relevant to the topic may have been missed, which may have caused some selection bias. On the other hand, it enabled a broader search so we could explore the recent themes of PV. It was also limited to citing the most frequent terms found, which do not exhaust the topic, as a range of terms that indicate ineffectiveness and inappropriate use can be found in the various PV databases according to the culture and training of the reporters.

Although the strategies were divided into two major topics (ineffectiveness and inappropriate use) to facilitate data organization, in reality, they are found together, all signalling problems related to ABs. PV studies are not definitive in their conclusions, as they are, by nature, about suspected AEs; however, they may indicate guidelines and paths to be followed and investigated to confirm possible AMR.

## 7. Materials and Methods

This narrative review was carried out by conducting searches in the Pubmed, Embase, and Lilacs databases between December 2023 and March 2024. The steps outlined in Figure 3 were followed to ensure the selection of relevant articles and the extraction of pertinent information.

## 8. Conclusions

This review showed strong support for the idea that PV studies may be useful in the use of ABs in addition to the already-known description of the safety profile of these medicines: (1) PV can help identify AB ineffectiveness, and (2) it can identify inappropriate use. This use of well-known tools already consolidated in many countries can help guide AMS and drug-related policies so that healthcare providers can be alerted about a possible lack of response due to AB resistance.

In a scenario where access to adequate microbiological diagnosis and monitoring is difficult, it is necessary to take advantage of every possible source of available information, such as PV programs and their databases. The studies presented in this review showed not only the presence but the frequent appearance of terms of interest about AMR on a local, national, and international basis. These strategies can be incorporated as indicators of inappropriate use and AB resistance in AMS. Knowing the roots of the causes of these types of events and encouraging specific notifications can make transformative contributions to stewardship programs.

The main aim of this review was to show the feasibility of using a well-known and consolidated public health tool to contribute to antimicrobial stewardship activities and to address any resistance problems.

From this perspective, strengthening PV systems and encouraging reporting of specific AEs related to AMR could be important indicators for monitoring the use of ABs and advancing AMR-related national policies and stewardship programs.

## Figures and Tables

**Figure 1 antibiotics-13-00457-f001:**
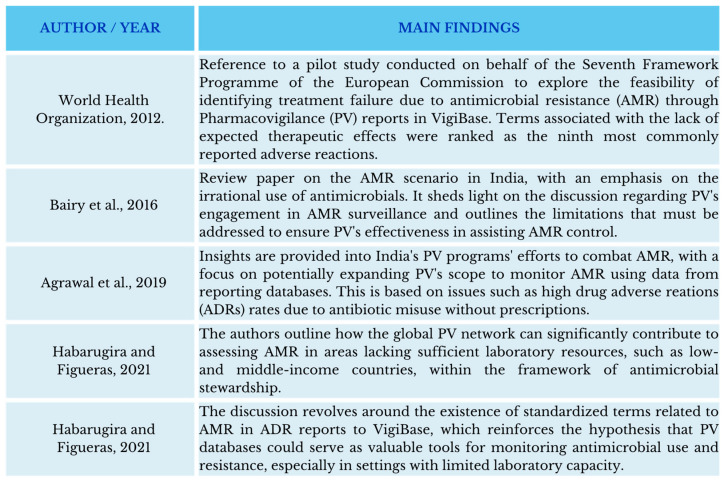
Evolution of publications in the literature regarding the involvement of pharmacovigilance in actions to monitor antimicrobial resistance [24,25,27,29,98].

**Figure 2 antibiotics-13-00457-f002:**
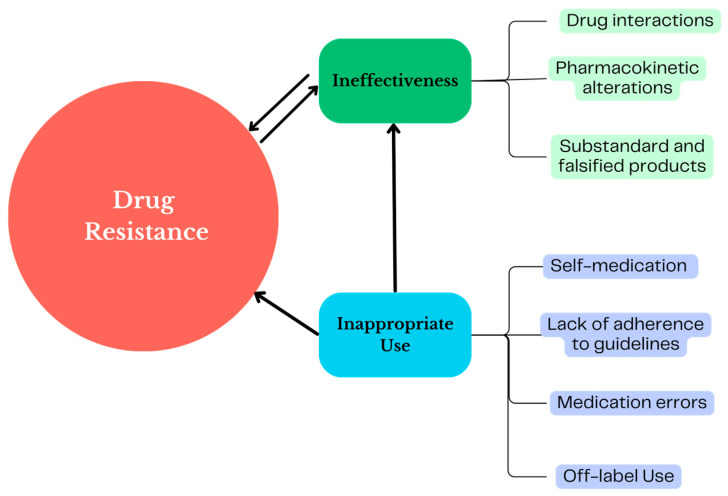
Scheme of real-life antibiotic use problems and interfaces with antimicrobial resistance that can be identified in pharmacovigilance studies.

**Figure 3 antibiotics-13-00457-f003:**
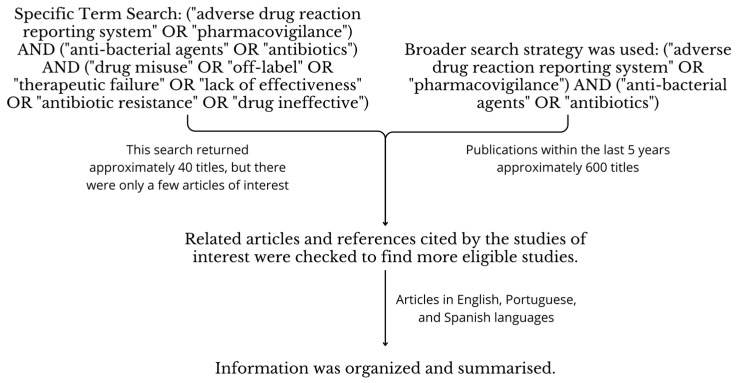
The steps outlined for the selection of relevant articles are included in this study.

**Table 1 antibiotics-13-00457-t001:** MedDRA terms related to effectiveness and inappropriate use and the AWaRe classification of antibiotics described in the synthesised studies.

MedDRA Terms		
Drug-ineffective		
Pathogen resistance		
Drug resistance		
Multidrug resistance		
Product use issue *		
Product use in inappropriate indication		
Incorrect route of administration		
Inappropriate prescribing *		
Indicating antibiotics for patients with no sign of infection *		
Inappropriate indication for antimicrobial prophylaxis *		
Inappropriate indication *		
Readministration of antibiotics causing prior allergy/allergies *		
Inappropriate dosage *		
Inappropriate dose and dosage regimen *		
Contraindicated administration *		
Late or irrational change of antibiotic in case of treatment failure *		
Administration of an inappropriate treatment regimen for the disease/Inadequate treatment strategy *		
Off-label use		
**AWaRe classification of the antibiotics included in the synthesized studies**		
**Access**	**Watch**	**Reserve**
Amikacin	Azithromycin	Aztreonam
Amoxicillin	Cefoperazone	Colistin
Amoxicillin/clavulanic acid	Ceftazidime	Daptomycin
Ampicillin	Ceftriaxone	Linezolid
Cefalotin	Cefuroxime	Polymyxin-B
Cefazolin	Ciprofloxacin	Tedizolid
Doxycycline	Clarithromycin	
	Levofloxacin	
	Meropenem	
	Piperacillin/tazobactam	
	Tobramycin	
	Vancomycin	
Ceftriaxone/tazobactam **		

BLUE: terms related to ineffectiveness. GREEN: terms related to inappropriate use. * Terms are described as they appear in the articles and may not correspond exactly to the term as listed in the MedDRA dictionary. ** The AWaRe classification for this fixed-dose combination is Not Recommended.

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
