# Peer review of "Pharmacovigilance Strategies to Address Resistance to Antibiotics and Inappropriate Use—A Narrative Review"

_antibiotics, 2024, doi:10.3390/antibiotics13050457_

Round 1
Reviewer 1 Report
Comments and Suggestions for Authors
Thank you for the opportunity to review this manuscript pertaining to the use of PV in the detection of AMR. This is a well thought-out concept and is pertinent to antimicrobial stewardship efforts to curb rising rates of antimicrobial resistance. I commend the authors for the work they have done and the summary of the available literature. I do have some outstanding questions regarding actionable items that may be taken from this work and an understanding as a whole.
1. One area that is addressed briefly is under-reporting of ADEs that may skew the data portrayed in this review. I think more commentary is needed on the importance of PV and complete reporting. Additionally, authors should comment on missing pieces of data or data that was incorrectly entered into PV surveys. I think in that while this manuscript offers some great thoughts, there are some practical limitations related to reporting of PV that may hinder implementation. As a whole, I think the limitations section should be expanded to include this information as well as that made in point 2.
2. One concern I have with rising rates of antimicrobial resistance is that we are too reactive. In other words, we find resistance and then adapt to it. As it is currently reviewed, this manuscript presents a way of potentially detecting resistance, but I would encourage the authors to think of ways PV could be used as a proactive way of preventing antimicrobial resistance. This could make for a great future directions and give readers actionable ways to implement PV in the prevention of AMR.
Comments on the Quality of English Language
As a whole, this manuscript is well-written and the quality of the English language is very good. There are a few instances of run-on sentences that should be addressed.
Reviewer 2 Report
Comments and Suggestions for Authors
Dear reviewers, I had the pleasure of reading your manuscript entitled "Pharmacovigilance strategies to address resistance to antibiotics and inappropriate use – A narrative review" I have some comments.
The manuscript can be enriched with a flow diagram of the literature review process.
- Some abbreviations, such as ADRs and AE, have not been defined.
- Figure 1: It is not clear; the arrows do not make sense, and the figure is not understood.
- Lines 146-147, I think that references to supplementary materials should not be included in the manuscript.
I hope that these comments will be helpful in improving the manuscript.
Reviewer 3 Report
Comments and Suggestions for Authors
Pharmacovigilance strategies to address resistance to antibiotics and inappropriate use – A narrative review
This manuscript underscores the potential utility of existing pharmacovigilance (PV) systems to contribute to Antimicrobial Stewardship (AMS) initiatives. Generally, The text is well-structured and presents a detailed overview of the role of pharmacovigilance (PV) systems in addressing antimicrobial resistance (AMR).
§ Line 60-70: Comparing the elevation in Reserve AB consumption in the Brazilian intensive care units during the COVID-19 pandemic should be against the same situation in high-income countries. Consider including more specific examples from different regions or countries to elaborate on how and why the pandemic led to an increase in antibiotic use. This trend seems to have been replicated globally, regardless of economic disparities.
§ Line 72-80: You introduced pharmacovigilance as a helpful strategy for tracking AMS. Consider highlighting some real-world examples that demonstrate the practical application and effectiveness of PV in healthcare settings, and how it complements the AMS pillars. Highlight briefly the specific benefits of using PV data, such as the early detection of AMR trends, identification of treatment failures, and reporting of inappropriate antibiotic use.
§ Line 87-100: The final paragraph of the introduction appears to be more of a methodology for the work. Therefore, it should be detailed under a distinct title. In the last paragraph, the focus should be on outlining the study objectives, limitations, and the novelty of the work.
§ Line 150-170: Several studies and references are cited in the paragraph, but it would be helpful to provide summaries, key findings, or brief definitions from these studies to support the claim to enhance the readability of the information presented.
§ Line 162-170: Briefly describe the terms utilized in both referenced studies to signify treatment failures caused by AMR reported by PV systems.
§ Line 201-264: According to your assertion, previous studies have identified particular terms in pharmacovigilance databases that indicate treatment efficacy, such as "drug ineffective," "treatment failure," "decreased drug efficacy," and "therapy nonresponder." It's important to note that these terms may not solely result from Antimicrobial Resistance (AMR); they can also stem from inappropriate antibiotic selection or prescription, and suboptimal regimen adjustments, among other factors. How can researchers address these confounding variables to isolate and accurately conclude unmet treatment outcomes linked specifically to AMR?
General
§ To enhance readers' convenience, I suggest adding a dedicated definition section to provide explanations for all terms mentioned throughout the manuscript. This will ensure clarity and facilitate better understanding.
§ Providing more detailed insights into how PV systems have been applied in real-world scenarios or the outcomes of certain interventions would enhance the credibility and relevance of the discussion.
§ While discussing the benefits of PV systems in addressing AMR, it's also imperative to identify any limitations or challenges associated with these systems.
